# Shape Deviation of Surface Structures Produced by WaveShape (Structuring by Laser Remelting) on Ti6Al4V and a Method for Deviation Reduction

**DOI:** 10.3390/mi12040367

**Published:** 2021-03-29

**Authors:** Oleg Oreshkin, Daniil Panov, Laura Kreinest, André Temmler, Alexander Platonov

**Affiliations:** 1Moscow Engineering Physics Institute, National Research Nuclear University MEPhI, Kashirskoe Shosse 31, 115409 Moscow, Russia; xxplatonov@ya.ru; 2Center for Design, Manufacturing & Materials, Skolkovo Institute of Science and Technology, Bolshoy Boulevard 30, bld. 1, 121205 Moscow, Russia; Daniil.Panov@skoltech.ru; 3Chair for Digital Additive Production, RWTH Aachen University, Campus-Boulevard 73, 52074 Aachen, Germany; laura.kreinest@dap.rwth-aachen.de; 4Fraunhofer Institute for Applied Optics and Precision Engineering, Albert-Einstein-Straße 7, 07745 Jena, Germany

**Keywords:** laser surface treatment, laser structuring, WaveShape, shape deviation, Ti6Al4V, laser processing, surface structuring

## Abstract

Laser structuring by remelting (WaveShape) is a manufacturing process for metal surfaces in which structures are generated without material removal. The structuring principle is based on the controlled motion of the three-phase line in the area of the solidification front. The contour of the solidification front is imprinted into the remelting track during the continuous solidification process. Typically, harmonic surface structures in the form of sinusoidal oscillations are generated by means of WaveShape with virtually no material loss. However, a significant shape deviation is often observed over a wide range of process parameters. In this study, it was found that much of the shape deviation is concentrated at a spatial wavelength equal to half the spatial wavelength used for structuring. Therefore, an approach to reduce the shape deviations was specifically investigated by superimposing a compensation signal on the harmonic structuring signal. In this approach, a compensation signal with half the spatial wavelength was varied in phase and amplitude and superimposed on the structuring signal. Amplitude and phase shift of the compensation signal were further investigated for selected laser beam diameters and spatial wavelengths. This demonstrated that a shape deviation of harmonic surface structures on titanium alloy Ti6Al4V could be reduced by up to 91% by means of an adapted compensation signal.

## 1. Introduction

Surface functionalization through surface topography adaption is applied in a variety of applications as, e.g., to control tribological properties [1], in biomedical applications [2], to improve wettability characteristics [3] or for bonding between different materials [4].

Functionalization of a surface consists in adapting the surface properties. New properties can be established by changing the chemical and phase composition of the surface, as well as changing the surface morphology.

Researchers have various surface treatment methods at their disposal to adapt the surface properties. They can be divided into low-scale methods, where high-selectivity of the process is required (e.g., micro-milling [5], electron beam processing [6], and various laser-based methods), and large-scale methods, where a significant part of the area can be treated in the uniform manner (for example, sandblasting [7], electrical discharge machining (EDM) [8], or electrochemical etching [9]). Despite a wide variety of processing purposes, surface treatment methods are mainly based on the removal of material up to predetermined spatial properties. Moreover, regular structures with repeating patterns are largely created by local processing methods.

Laser-based material structuring technologies are often associated with laser ablation. This approach requires high intensity laser radiation provided by pulsed laser sources. The spatial sizes of the structures produced by ablation can vary from submicron structures created with DLIP (Dual Laser Interference Patterning) [10,11] or LIPSS (Laser-Induced Periodic Surface Structures) [12] technologies to relatively large sizes using LST (Laser Surface Texturing) [13], as well as scanning strategies, over a large area [14].

## 2. State of the Art

A different approach to surface topography adaption is laser remelting technologies. These processes generally require less power density for the absence or a low rate of vaporisation. Laser polishing is a remelting-based process designed to remove surface roughness. Depending on the initial surface and the material, both continuous wave (CW) and pulsed laser sources can be used [15]. Furthermore, the remelting approach can be applied to various materials. Weingarten [16] applied it to remove roughness from glass, in Reference [17] remelting was applied to a ceramic material for roughness reduction. By References [18,19,20], laser polishing was applied for treatment of steel and titanium.

The structuring technology WaveShape (see Figure 1) is based on laser remelting and was first introduced by Temmler [21]. During this process, regular surface structures are generated by the redistribution of material in the molten state. The resulting structure height is linearly dependent on the laser power amplitude and can be adjusted from a few microns up to several hundreds microns.

By laser processing with the beam diameter $d_{L}$, a molten pool is created on the metal surface, which moves along the scanning direction with a scan speed $v_{scan}$. The scanning speed is generally varied from 10 to 200 mm/s. Using constant laser power $P_{M}$ in the order of a few hundred watts, the direction of solidification is parallel to the initial surface in the direction of the beam movement. However, a change of the laser power leads to a change of the direction of the solidification front following the change of the molten pool size. Therefore, when the melt pool volume increases, the direction of solidification shifts upwards and when the volume of the molten pool decreases, it moves downwards [23].

During solidification, a texture is formed on the surface which has one to one correspondence to the laser power modulation signal. In order to obtain harmonic structures, the average laser power $P_{M}$ is modulated during the process with a laser power amplitude $P_{A}$ in the order of several tens of watts at a predetermined spatial wavelength λ.

The nature of the process lies in the behavior of the molten pool driving forces. For the WaveShape process, Temmler [22] considers spontaneous shape change due to volume expansion as a main principle of the structuring. Sharma [24] provides a 2D-process simulation of the WaveShape process by COMSOL Multiphysics and considers the Marangoni force as the main force.

The technology can be applied to adapt the surface morphology. According to Bordatchev [25], potential applications of the technology are the structuring of moulds for the production of light guides and texturing metal surfaces. An approach for removing waviness was considered by Oreshkin Reference [26]; the principle is called destructive interference.

This paper focuses on the creation of harmonic structures, although the generation of structures with complex shapes using WaveShape is a promising area of research.

## 3. Problem Statement

Surface structures created with the laser structuring by remelting normally have a one to one correspondence of the resulting surface structure and the modulation signal. However, even for harmonic structures, shape deviations can be observed. When the power modulation signal is, e.g., given as $P\left(x\right)=P_{M}+P_{A}\cdot cos\left(\frac{2\pi x}{\lambda }\right)$, the resulting surface profile can mathematically be described as:(1)$$H\left(x\right)\propto P_{A}\cdot cos\left(\frac{2\pi x}{\lambda }+\phi \right)+\sigma \left(P_{A},\lambda \right),$$
where ϕ is the phase shift (see Figure 2), $\sigma (P_{A},\lambda )$ is a deviation component, presumably depending on the laser power amplitude $P_{A}$ and the spatial wavelength λ.

Temmler [23] considered this issue and divided it into two parts. The first is the phase shift between the modulation signal and the generated structures ϕ presented in Figure 2, the second is the so-called asymmetry. The asymmetry of the structures was measured as the asymmetry angle θ:(2)$$\theta =arctan(\Delta \lambda /h_{max}),$$
where Δλ is the peak deviation from the symmetry position in the lateral direction, and $h_{max}$ is the height amplitude of the structure (peak). An example of structures with asymmetry is displayed in Figure 3.

The approach of Temmler [23] to reduce asymmetry is the double structuring of a track: first in one direction and then in the opposite direction, so that the deviation caused by the first pass is compensated. This method can reduce the shape deviation but requires twice as much processing time.

It might be more accurate to describe the shape deviation of structures rather than their asymmetry because, although the structures on the surface might be symmetrical, they could still exhibit a deviation from a predetermined shape.

Shape deviation is a key factor for the inverse laser structuring problem: the generation of a laser power modulation signal to generate a predetermined structure on the surface. Thus, if the power modulation signal does not allow the generation of a given structure, it is necessary to modify the modulation signal (see Figure 4). This paper describes research towards finding such a modulation signal modification.

## 4. Materials and Methods

### 4.1. Materials and Surface Preparation

Commercially available Ti6Al4V alloy was used as a base material for this research. The probes with flat surface were mechanically grinded with grinding wheels up to grit size P500 according to ISO-6344. The measured residual arithmetical mean roughness $R_{a}$ after probe preparation was lower than 0.1 μm. Before laser processing, the surface was cleaned with isopropanol.

### 4.2. Experimental Set-Up

The experimental set-up is presented on the Figure 5.

In the scope of this study, the laser processing was done with the CW fiber laser IPG LS-2 (IPG Photonics, Fryasino, Russia, legacy model of YLS-2000) with maximum laser power up to 2 kW. In addition, a 3D-Laser scanner (IPG Photonics, Oxford, MA, USA) was used. The scanning speed was fixed on $v_{scan}$ = 25 mm/s. The power modulation signal was created with a Control Cell (Blackbird Robotersysteme GmbH, Garching—Hochbrück, Germany). An industrial robot ABB IRB 4600 (ABB, Zürich, Switzerland) was used for the alignment of the laser scanner. The process gas chamber was filled with inert gas (Argon) to a residual oxygen concentration of 1000 ppm. Glass with anti-reflection coating was used to reduce back reflection. The oxygen concentration during the processing was measured with the gas analyzer AKPM-1-01 (Alfa Bassens, Moscow, Russia).

The structures were obtained by scanning single tracks with a length of six times the spatial wavelength but no less than 20 mm. The power modulation was provided as a periodic function:(3)$$P\left(x\right)=P_{M}+P_{A}\cdot cos\left(\frac{2\pi x}{\lambda }\right),$$
with $P_{M}$ defined as
(4)$$P_{M}=\frac{P_{MAX}+P_{MIN}}{2}$$
and $P_{A}$ defined as
(5)$$P_{A}=0.8\frac{P_{MAX}-P_{MIN}}{2},$$
where the $P_{MIN}$ is the laser power at which the melting process starts, and $P_{MAX}$ the laser power at which a plasma torch is visible above the molten pool. The laser power amplitude $P_{A}$ was calculated with a coefficient of 0.8 to remain within the safe limits of the processing window.

Table 1 shows the laser power parameters which are determined for the scanning velocity $v_{scan}$ = 25 mm/s and laser beam diameters $d_{L}$ = 250 μm and $d_{L}$ = 500 μm.

### 4.3. Surface Measurement and Data Processing

For the measurement of the longitudinal profiles of the generated structures, a Dektak 150 stylus profilometer (Veeco Instruments Inc., Plainview, NY, USA) was used.

The result of the measurement is a two-dimensional structure profile h(x). A Fourier transformation F[h(x)] was performed on the profile with rectangular windowing. The width of the rectangle window can be determined from the analysis of the fundamental frequency of the obtained profiles. The following parameters was determined:Power spectrum magnitude |F(f)| on the first spatial frequency ($f_{1}$) and the second spatial frequency ($f_{2}$). Further, in order to avoid confusion, the term “harmonic” is used to indicate certain frequencies or wavelengths. The first spatial frequency $f_{1}$ (first harmonic) corresponds to the main spatial wavelength of the power modulation signal P(x); the second harmonic is $f_{2}=2f_{1}$. The spatial frequency is inverse to the spatial wavelength (λ=1/f).The ratio of the magnitudes on the second and first harmonics $R_{2/1}$. The ratio of harmonics is used as a measure of structure deviation.The phase shift of the second harmonic relative to the first harmonic Δϕ is measured as: $\Delta \phi =\phi \left(f_{2}\right)-2\phi \left(f_{1}\right)$, where $\phi (f_{1})$—shift of the first harmonic, and $\phi (f_{2})$—shift of the second harmonic.

## 5. Results and Discussion

### 5.1. Determination of Process Parameter Sets

In order to investigate different types of deviations, laser processing was done for two fixed beam diameters with a variation of the spatial wavelength. For selected spatial wavelengths and beam diameters, the power modulation amplitude $P_{A}$ is varied in five steps to the maximal $P_{A}$ measured during preliminary experiments: for $d_{L}$ = 250 μm, $P_{A}$ is varied from 20 to 52 W; for $d_{L}$ = 500 μm, $P_{A}$ is varied from 30 to 176 W. The magnitudes of the second and first harmonics, as well as their ratio as a function of the spatial wavelength, is presented in Figure 6. For both beam diameters, the maximum structure height of the first harmonic is reached at a spatial wavelength of approximately four times the beam diameter. This dependence is close results observed by Temmler [23]. However, additional analysis was done to examine the deviation of the signal from the power modulation signal.

The minimum of the ratio $R_{2/1}$ can be determined to $\lambda =4d_{L}$ for a beam diameter of $d_{L}$ = 250 μm and $\lambda =4.5d_{L}$ for a beam diameter of $d_{L}$ = 500 μm, respectively. Thus, the symmetric structures with low shape deviation can only be generated in a small region of wavelengths. For regions of small and large wavelengths, the shape of the structure differs from the symmetric structure. Furthermore, general structure heights are lower for these ranges. One approach to adapt the shape of the structures in regions of small and large wavelengths to the symmetric target shape was investigated in this study.

Three regions of wavelengths were determined: a small wavelength region (1–2$d_{L}$), a medium wavelength region (3–5$d_{L}$) and a large wavelength region (>5$d_{L}$). For the following experiments, one main spatial wavelength λ1 from each region is selected (see Table 2).

Temmler [27] investigated structuring by remelting for the Ti6Al4V alloy. The structure height was used as a parameter of the processing quality. For $v_{scan}$ = 25 mm/s, structure heights of up to 20 μm for a beam diameter of $d_{L}$ = 250 μm and up to 40 μm for a beam diameter $d_{L}$ = 500 μm were generated. The maximum of the structure height is also observed in the region of medium spatial wavelengths. The deviation of the structure shape has not been investigated. In this study, the processing is carried out with lower power density: for a beam diameter $d_{L}$ = 250 μm, $P_{M}$ = 92.5 W and $P_{A}$ = 52.5 W were determined, and, for a beam diameter $d_{L}$ = 500 μm, $P_{M}$ = 130 W and $P_{M}$ = 105 W were determined. Presumably, this inconsistency is caused by the different distribution of the power density in the in the beam spot.

The nature of the structure height distribution of the as a function of the ratio $d_{L}/\lambda$ is consistent with earlier studies of both the titanium alloy Ti6Al4V [27] and steel H11 [23]. The existence of an extremum in the medium wavelength region can presumably be explained by the coordinated motion of the three-phase line of the solidification front with the expansion-compression cycle of the molten pool during structuring by remelting. Further research is needed to confirm or refute this hypothesis.

Furthermore, it can be observed that the range of the ratios $R_{2/1}$ for different beam diameters is the same for different magnitudes of the first harmonics (see Figure 6). This indicates that the ratio $R_{2/1}$ is a suitable measure to compare the quality of different harmonic structures.

### 5.2. Linearity of Laser Power Amplitude on Structure Shape Deviation

It is known that the structure heights linearly increase with the increase of laser power amplitude $P_{A}$ [23] for various materials. In this study the influence of the laser power amplitude $P_{A}$ to structure shape deviation was investigated. For selected spatial wavelengths and beam diameters, the laser power amplitude $P_{A}$ was varied in five steps to the maximal $P_{A}$ measured during preliminary experiments: for $d_{L}$ = 250 μm, $P_{A}$ was varied from 20 to 52 W; for $d_{L}$ = 500 μm $P_{A}$ was varied from 30 to 176 W. The ratio of the second and first harmonics is shown in Figure 7.

The tendency for structuring with lower shape deviation in the medium wavelength range is more noticeable at higher $P_{A}$. The reason is presumably the low amplitude of the first harmonic, since as a consequence even a small variation of the amplitude results in an increase of the ratio of harmonics. The graphs in Figure 7 show the predominant trend of an increase of $R_{2/1}$ and, therefore, shape deviations with an increase of the laser power amplitude $P_{A}$ excluding the lowest values. A presumable reason might be an increase in the unevenness of the three-phase line movement at higher laser power gradients $dP_{A}\left(x\right)/dx$.

#### 5.2.1. Direction of Phase Shift

For chosen spatial wavelengths and laser power parameters described in Section 5.1, the phase shift of the second harmonic relative the first harmonic Δϕ were measured (see the Figure 8).

For small spatial wavelengths Δϕ > 0°, this indicates that structure peaks are shifted to the left from the symmetrical position. For the large spatial wavelengths Δϕ < 0°, the peak is shifted to the right from the symmetrical position. These findings are consistent with the data obtained on the steel 1.2343 [23]. For the medium spatial wavelengths, the spatial shifts are located between the extreme values. The sign of the phase shift Δϕ in both the low wavelength and large wavelength range is consistent with the previous investigation of shape deviation on steel 1.2343.

#### 5.2.2. Compensation of Laser Power Amplitude

In order to improve the structure quality by reducing the surface shape deviation, a compensation signal has been added to the signal of laser power amplitude. Since the majority of the deviation is caused by the second harmonic, the compensation signal is a harmonic wave with the doubled frequency with a phase shift relative to the first harmonic. As a result, the Formula (Equation 3) of the laser power signal changes to the following:(6)$$P\left(x\right)=P_{M}+P_{A}\cos(\frac{2\pi }{\lambda_{1}}x)+P_{A2}cos(\frac{2\pi }{\lambda_{1}/2}x+\varphi_{2}).$$

The laser power amplitude of the compensation signal was chosen based on the ratio $R_{2/1}$ for the corresponding laser beam diameter and spatial wavelength:(7)$$P_{A2}=R_{2/1}P_{A}.$$

The process parameters for every parameter set are shown in Table 3.

For each parameter set, the phase shift $\phi_{2}$ is varied from −180° to 180°. The results of the structuring with the compensation signal are shown in Figure 9. The influence of the compensation signal is compared to the reference structure with $P_{A2}$ = 0 W.

From the graphs in Figure 9, it can be seen that adding the compensation signal to the laser modulation signal has an influence on the structure shape deviation. For each parameter, there is a value of the phase shift $\phi_{2}$, where the ratio of harmonics is minimal. For small spatial wavelengths, the minimal value is in the range of −30° to 60°; for large spatial wavelengths, this range is approximately 120° to 150°. For the medium spatial wavelengths, no trend in the location of the extremum could be determined as general for both beam diameters. Additionally, the magnitude of $R_{2/1}$ is much lower in the medium spatial wavelength range, so the reference structures are more symmetrical than they are in the other ranges. Thus, the influence of phase shift on shape deviation in the medium spatial wavelength range is lower than in the other ranges.

Moreover, the graphs in Figure 10 show the correlation between the phase shift of the structures obtained with the compensation signal and the position of the ratio extrema. The phase shift of the structure obtained with the compensation signal corresponds to the phase shift of the reference signal at the points of the extrema. This implies that the phase shift of the second harmonic does not change at the extreme points. For the maximum of the ratio of harmonics $R_{2/1}$, a positive interference occurred, and, for the minimum of the ratio of harmonics $R_{2/1}$, a negative interference appeared.

#### 5.2.3. Finding the Optimal Value of Laser Power Amplitude

After finding the extreme points by variation of the phase shift $\phi_{2}$, the investigations of the amplitude of the compensation signal is performed. The process parameters from the Table 3 are used, but the laser power amplitude of the compensation signal $P_{A}$ is varied in twelve steps. The results of the investigations are displayed in Figure 11.

For each parameter set, the dependence of the ratio of harmonics $R_{2/1}$ on the laser power amplitude of the compensation signal $P_{A2}$ is investigated. The local minima show the optimum value of the amplitude $P_{A2}$ for a given phase shift $\phi_{2}$. A further increase of the laser power amplitude will lead to a decrease in the structure quality due to overstructuring at the second harmonic. As a result, Table 4 shows the parameter sets for each beam diameter $d_{L}$ and spatial wavelength λ, where the harmonics ratio is minimal. The harmonics ratio for the reference structure and the structure after processing with the compensation signal is also presented.

Examples of the reference structure and the structure after processing with compensation are shown in Figure 12 for comparison. For the structures in the small wavelength range, the peak of the reference structure is shifted to the left. For the large wavelength range, the peak of the structure is shifted to the right. After processing with the compensation signal, both structures show no visible signs of a shift.

## 6. Summary and Outlook

Laser structuring by remelting (WaveShape) enables the generation of harmonic structures on metal surfaces. The structures are formed due to the kinematics of the three-phase line of the melt pool. In this study, the maximum structure heights were achieved at $d_{L}$= 4λ for $d_{L}$= 250 and 500 μm, which is in good agreement with other studies. Additionally, shape deviations of the generated sinusoidal structure were systematically investigated for the first time.

It was found that the majority of the shape deviation are concentrated at a spatial wavelength corresponding to half the spatial wavelength of the structuring. Therefore, shape deviation was estimated using, inter alia, the amplitude ratio of the first harmonic (structuring wavelength) and second harmonic (half structuring wavelength) $R_{2/1}$. It was observed that the shape deviations for medium spatial wavelengths (λ = 3–5$d_{L}$) are significantly smaller than for small (λ = 1–2$d_{L}$) or large (λ > 5$d_{L}$) spatial wavelengths. Thus, at wavelengths in the medium wavelength range, not only the highest structure heights but also the smallest shape deviations are obtained. However, a linear increase in the amplitude ratio $R_{2/1}$ (i.e., shape deviation) was observed with increasing laser power amplitude $P_{A}$, especially for longer spatial wavelengths. In general, the larger the laser power amplitude $P_{A}$, the higher the structural heights achieved. Therefore, reducing the shape deviation at large laser power amplitudes $P_{A}$ is of particular importance.

Therefore, in this study, an approach to reduce the shape deviation by superimposing a compensation signal on the harmonic structuring signal was specifically investigated. In this approach, a compensation signal with half the spatial wavelength (λ/2) was varied in phase shift ($\phi_{2}$) and amplitude ($P_{A2}$) and superimposed on the structuring signal. For all parameter combinations investigated, superposition with the compensation signal leads to a change in the amplitude ratio $R_{2/1}$, which depends, in particular, on the phase shift $\phi_{2}$. Thereby, the minimum of the amplitude ratio $R_{2/1}$ lies at a phase shift in the range of $\phi_{2}$ = −30° to −60° for small spatial wavelengths, and in the range of $\phi_{2}$ = 120° to 150° for large spatial wavelengths. Furthermore, the phase shift extremes of the second harmonic coincide with the phase shift of the reference signal. This indicates effective interference between the compensation signal and the processes that significantly affect the shape deviation. Additional adjustment of the power amplitude $P_{A2}$ reduced the overall $R_{2/1}$ amplitude ratio by up to 91%, resulting in a significant reduction of the shape deviation.

In summary, this work has demonstrated that shape deviation in the WaveShape process can be effectively corrected over a wide range of spatial wavelengths (λ = 2–8$d_{L}$). The correction is achieved by superimposing a wavelength-, amplitude-, and phase-corrected compensation signal on the structuring signal. An application of this approach could particularly improve the quality when generating more complex regular structures. Moreover, the implementation of compensation signal can extend the process limits to a wide range of spatial wavelengths without the need to change the laser beam diameter. Further planned investigations of shape deviation correction have the goal to realize a transfer from single tracks to surface processing (considering the track overlap).

Within the scope of this research, shape deviation was mainly considered from a signal processing perspective. The development of effectively adapted compensation algorithms will require a study of the physical correlations, especially taking into account the thermal properties of the material. In addition, this approach will be used, among others, for an active elimination of ripples and waviness on a metal surface.

## Figures and Tables

**Figure 1 micromachines-12-00367-f001:**
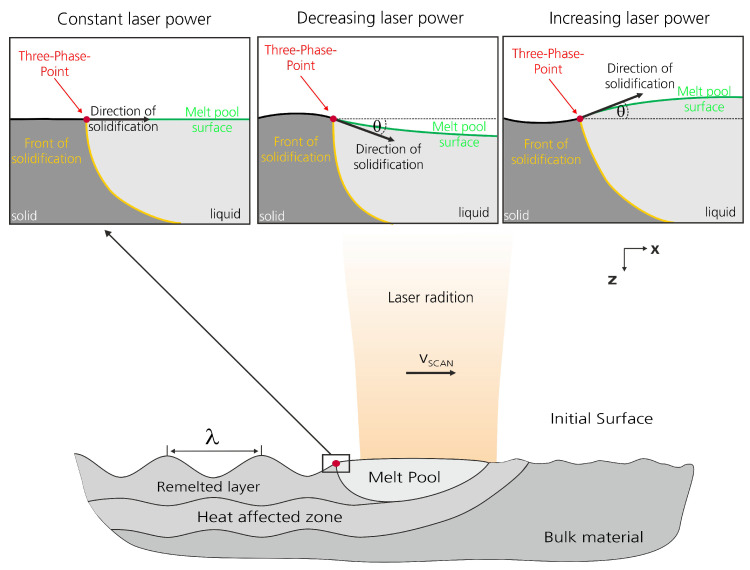
Schematic diagram of laser structuring by remelting (WaveShape) [22].

**Figure 2 micromachines-12-00367-f002:**
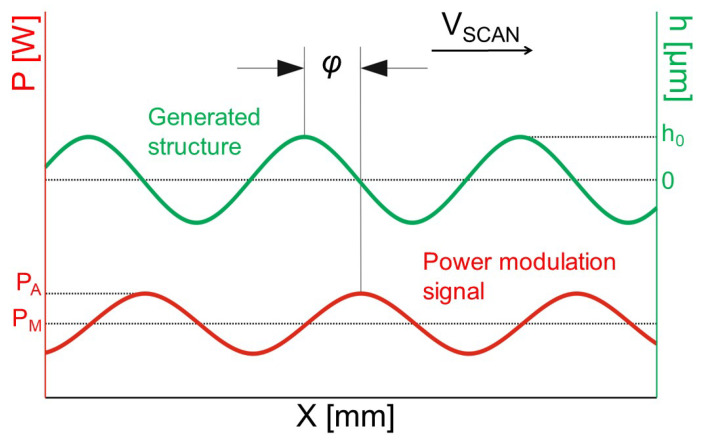
Schematic diagram of the relative position of the laser power amplitude signal P(x) and generated structure h(x) along the *x*-axis.

**Figure 3 micromachines-12-00367-f003:**
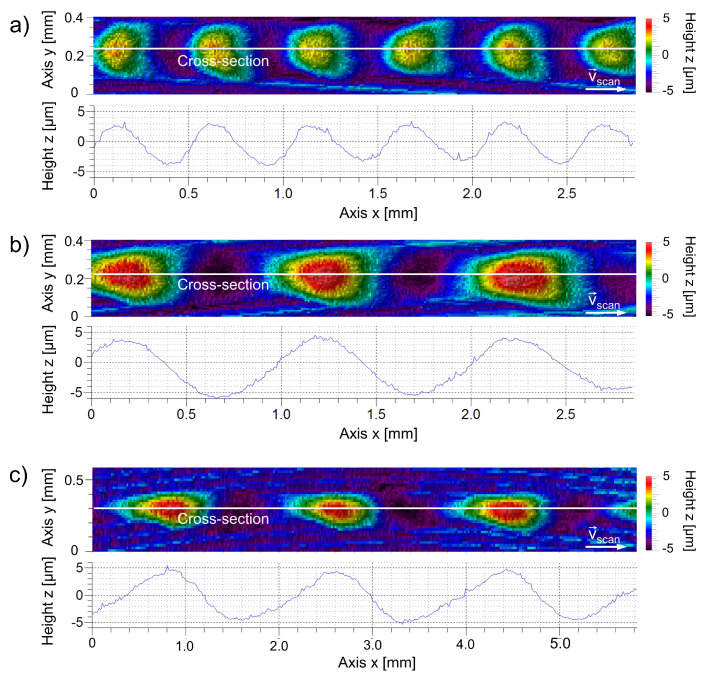
Single track structures on steel 1.2343 with: asymmetry of peaks shifted to the left (**a**), symmetrical in longitudinal direction (**b**), asymmetry of peaks shifted to the right (**c**); the 3D-profiles generated with a white light interferometer (1) and cross-sections along the longitudinal profile (2). Processing parameters: $P_{M}$ = 115 W, $P_{A}$ = 50 W, $d_{L}$ = 250 μm, $v_{scan}$ = 50 mm/s, (**a**) λ = 0.5 mm, (**b**) λ = 1.0 mm, (**c**) λ = 2.0 mm [23].

**Figure 4 micromachines-12-00367-f004:**
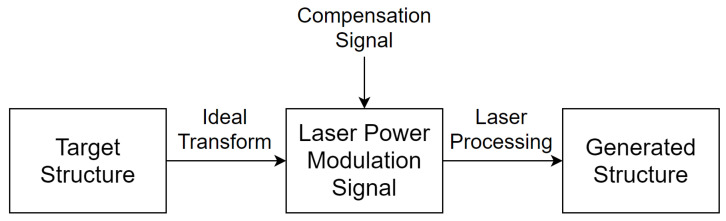
Schematic diagram of the Waveshape process chain with signal compensation.

**Figure 5 micromachines-12-00367-f005:**
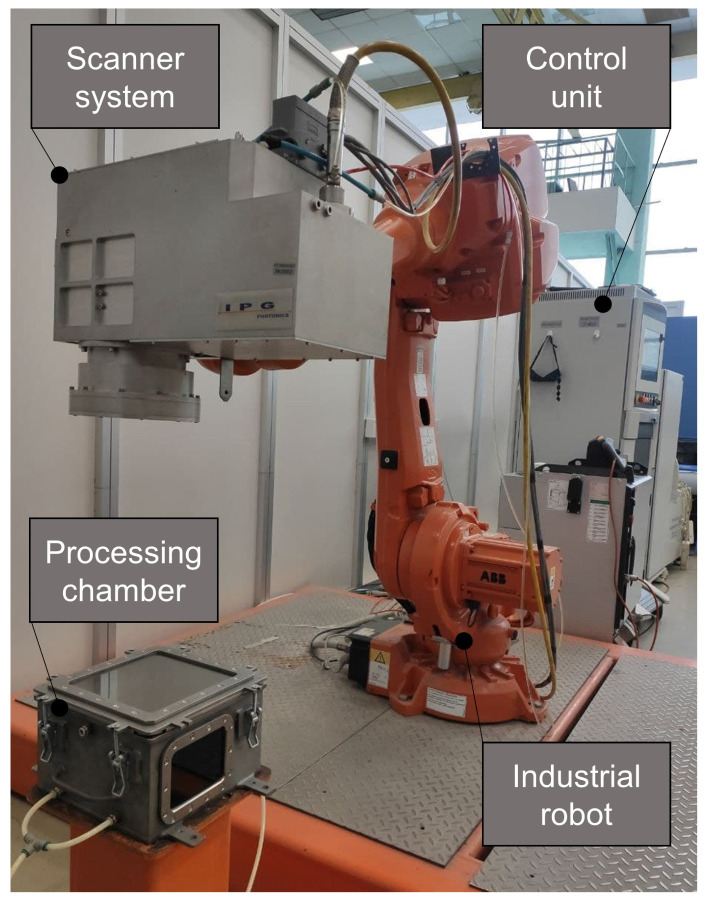
Experimental set-up for laser surface processing with process gas chamber for the avoiding oxidation of samples.

**Figure 6 micromachines-12-00367-f006:**
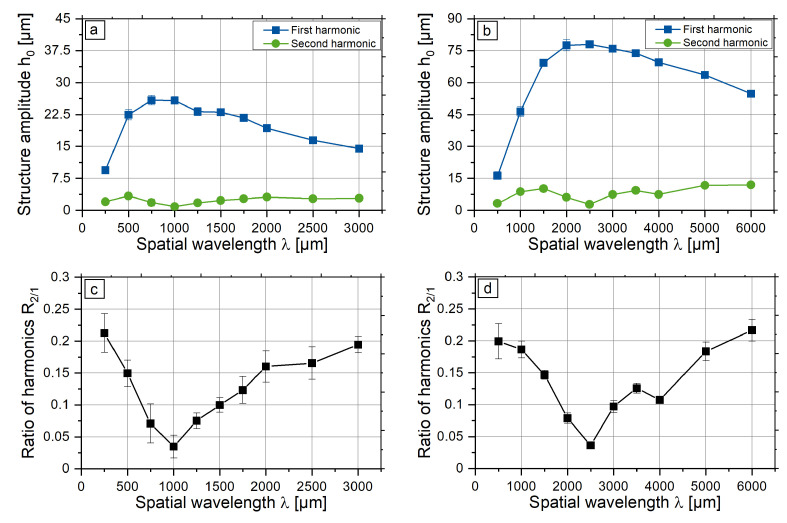
Structure height of the first and second harmonics for $d_{L}$= 250 μm (**a**) and 500 μm (**c**) and the ratio of the second and first harmonics $R_{2/1}$ (**b**,**d**), respectively.

**Figure 7 micromachines-12-00367-f007:**
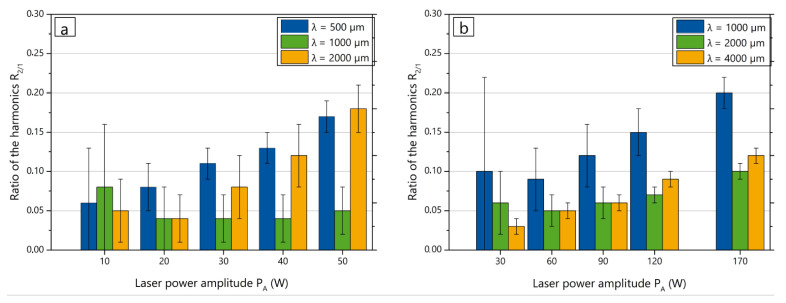
Ratio of the second and first harmonics $R_{2/1}$ in dependence of the amplitude of laser power modulation: (**a**) $d_{L}$ = 250 μm, (**b**) $d_{L}$ = 500 μm.

**Figure 8 micromachines-12-00367-f008:**
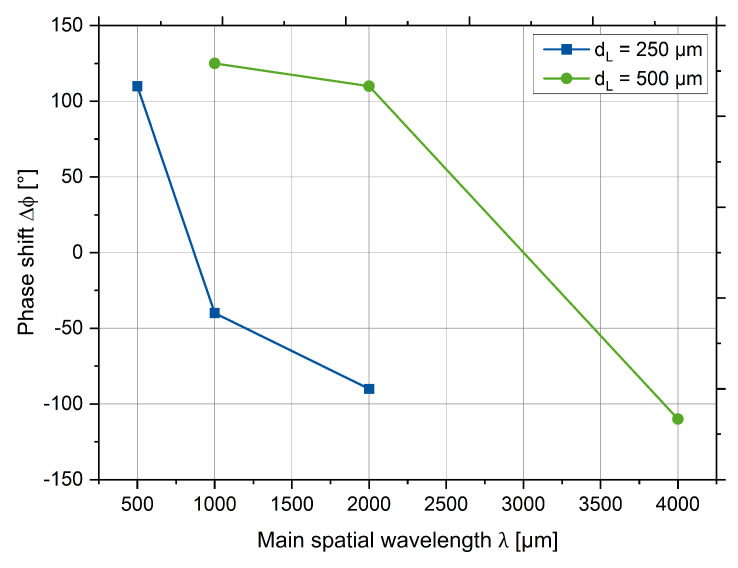
The phase shifts of reference structures.

**Figure 9 micromachines-12-00367-f009:**
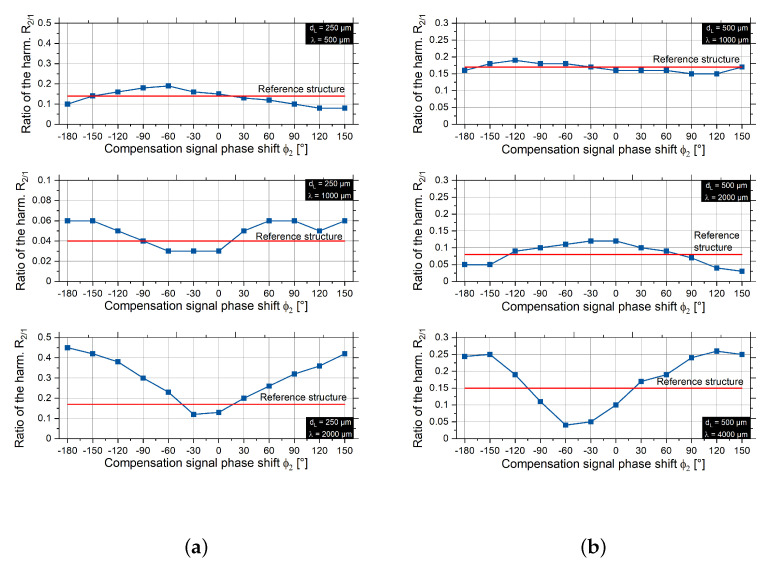
Ratio of the second and first harmonics $R_{2/1}$ depending on the phase shift $\phi_{2}$ of the compensation signal for $d_{L}$ = 250 μm (**a**) and 500 μm (**b**).

**Figure 10 micromachines-12-00367-f010:**
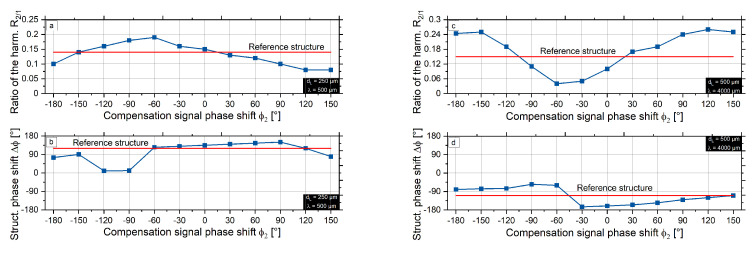
Phase shift accordance to the extremum of the harmonics ratio (**a**,**b**) $d_{L}$ = 250 μm, λ = 500 μm, (**c**,**d**) $d_{L}$ = 500 μm, λ = 4000 μm.

**Figure 11 micromachines-12-00367-f011:**
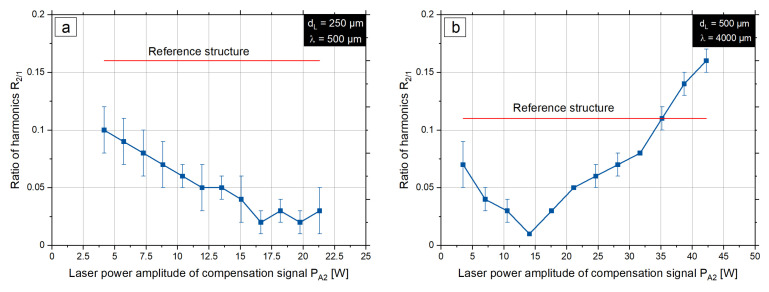
Ratio of the second and first harmonics $R_{2/1}$ depending on the laser power amplitude of the compensation signal: (**a**): $d_{L}$ = 250 μm, λ = 500 μm, (**b**): $d_{L}$ = 500 μm, λ = 4000 μm.

**Figure 12 micromachines-12-00367-f012:**
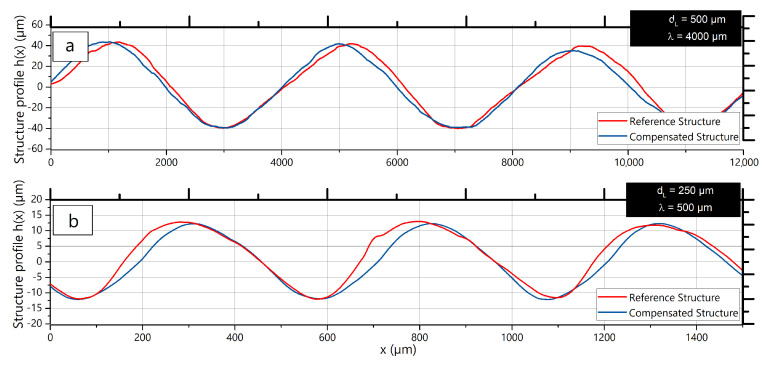
Reference and corrected surfaces: (**a**): $d_{L}$ = 250 μm, λ = 500 μm, (**b**): $d_{L}$ = 500 μm, λ = 4000 μm.

**Table 1 micromachines-12-00367-t001:** Process parameters for single tracks.

$d_{L}\left[\mu m\right]$	$v_{\mathrm{scan}}\left[\mathrm{mm}/s\right]$	$P_{M}\left[W\right]$	$P_{A}\left[W\right]$
250	25	135	52
500	25	410	176

**Table 2 micromachines-12-00367-t002:** Chosen spatial wavelengths from the small ($\lambda_{s1}$), medium ($\lambda_{m1}$), and large ($\lambda_{l1}$) wavelength regions.

$d_{L}$ [μm]	$\lambda_{s1}$ [μm]	$\lambda_{m1}$ [μm]	$\lambda_{l1}$ [μm]
250	500	1000	2000
500	1000	2000	4000

**Table 3 micromachines-12-00367-t003:** Process parameter sets for processing with the compensation signal.

$d_{L}$ [μm]	λ [μm]	$v_{\mathrm{scan}}$ [mm/s]	$P_{M}$ [W]	$P_{A}$ [W]	$P_{A2}$ [W]	$R_{2/1}$
250	500	25	135	52	8.84	0.17
250	1000	25	135	52	2.6	0.05
250	2000	25	135	52	10.92	0.21
500	1000	25	410	176	21.12	0.12
500	2000	25	410	176	17.6	0.10
500	4000	25	410	176	21.12	0.12

**Table 4 micromachines-12-00367-t004:** Process parameter sets for laser processing with optimized parameters of the compensation signal.

$d_{L}$, [μm]	λ, [μm]	$v_{\mathrm{scan}}$, [mm/s]	$P_{M}$, [W]	$P_{A}$, [W]	$P_{A2}$, [W]	$\phi_{2}$, [°]	$R_{2/1\mathrm{ref}}$	$R_{2/1\mathrm{comp}}$
250	500	25	135	52	16.6	–50	0.17	0.02
250	1000	25	135	52	0.26	160	0.05	0.03
250	2000	25	135	52	5.2	160	0.21	0.02
500	1000	25	410	176	72.6	–50	0.12	0.09
500	2000	25	410	176	17.6	–110	0.10	0.08
500	4000	25	410	176	14.4	140	0.12	0.01

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
