# Peer review of "Shape Deviation of Surface Structures Produced by WaveShape (Structuring by Laser Remelting) on Ti6Al4V and a Method for Deviation Reduction"

_micromachines, 2021, doi:10.3390/mi12040367_

Round 1

Reviewer 1 Report

This manuscript presents a novel method for the modification of metallic surfaces with highly controlled roughness via laser power modulation during beam scanning. The authors present the experimental method and results extremely well. The effect of scanning parameters including periodic power modulation is discussed as is the surface feature sizes that were capable of being created. This work would be of interest to readers working with laser material processing or other surface modification techniques, or in application areas where highly defined surface roughness is required.

One significant problem with the manuscript is the rather high level of self-citation with 8 out of 29 references [21-26, 28-29] citing author A. Tremmler. While it is understandable to self-cite especially where this is building on previous work, in my opinion this is higher than would be usual or acceptable. Some of the works cited are not recent (2009 and 2012). Many of these are referenced together for single statements for which one citation would suffice, while [22] is cited 7 times. In this reviewer’s opinion, this represents inappropriate self-citation and must be corrected before publication.

Pending the correction of this issue, plus the comments below, I would recommend this for publication.

Specific Comments:

Section 1/2 Introduction/State of the Art – Where previous work on the remelting approach is mentioned, can the authors provide more information on what lasers/powers/setups were used, along with what morphology/size features were produced? This will help contextualise the coming results in Section 3/4.

Section 5.1 - Can the authors provide more context in terms of what previous works have been able to achieve in terms of height/structure amplitude to allow the reader to contextualise the produced structures?

Line 26-27: “laser surface processing has the advantages of great controllability, high repeatability of results, lack of consumables and environmental friendliness”. That is an extremely broad statement and is not true for all laser processes, or all materials being processed. Indeed the author’s system requires both compressed air and water-cooling which are classed as “consumable”, and their process utilises an argon gas atmosphere. Please clarify this statement.

Line 37: “These technological processes generally require less intensity due to absent

38 or a low rate of vaporization.” Melting processes generally use lower irradiances or fluences, but often much higher average output laser powers than ablative surface machining laser processes. Please clarify statement

Line 69: “Problem Statement” or something similar would be a better title for this section.

Line 80: Can the authors comment on the two-dimensional asymmetry of the produced features? It seems that even when longitudinally symmetrical features are formed, there is asymmetry in the y-axis (as in Figure 3b1 and 3c1).

Line 103: I assume the authors are using an “LSS-2” from IPG not a “LS-2” which I could not find information on.

Line 179 “It applieds that for small spatial wavelengths Df>0°. This indicates that structure peaks are shifted to the left from the symmetrical position. It applies that for the large spatial wavelengths Df<0° and the peak shift is to the right. These findings are consistent with the data obtained on the steel 1.2343 5.1”. These sentences are unclear, please clarify

Line 184: “Adding the compensation part to laser power amplitude” I would retitle this section “Laser Power Amplitude Compensation” or something else clearer.

Author Response

Dear Reviewer,

Sincerely Yours

Dr. Oleg Oreshkin

Reviewer 2 Report

Dear Authors,

The topic looks interesting and the manuscript well organized, however following the below recommendation would improve the article to fir the Journal criterias:

1) Introduction is up to date, following the recent findings in this field but still needs improvement to cover all related aspects to the topic.

2) Figure 5: It is better to put a real photo from the experimental setup beside the schematic one

3) The results discussed should discuss in a quantitive way as well as the scientific way however a lack of deep discussion is clear. For such an article, it necessary to describe all the phenomena and happenings as well as reasons more in detail, compared to those previous outcomes from other researchers.

4) Conclusion may also be revised regarding the previous comment.

5) English should check by a native Technical English speaker.

Author Response

(The authors gave the same response as above.)

Reviewer 3 Report

The paper concerns on laser structuring of the Ti6Al4V titanium alloy surface by remelting (WaveShape).

The authors investigated the effect of laser processing parameters on reducing surface shape deviations after laser structuring.

The paper is very original and very interesting.

The paper was written correctly.

The language of the paper is satisfactory.
The authors drew proper conclusions.
The paper meets the journal's requirements and can be published.

Author Response

Dear Reviewer,
thank you very much for high estimation of our study. We did not  nd in
your comments the remarks for changing the manuscript.
Nevertheless, you can read the highlighted changes of the manuscript in the
new revision.
Yours Sincerely,
Oleg Oreshkin

Round 2

Reviewer 1 Report

The authors have addressed all the concerns raised in the review, and I recommend this article for publication. I wish the authors all the best in their future work.